# TEMPORALLY ALIGNED RELATION MODELING FOR PANOPTIC VIDEO SCENE GRAPH GENERATION

## ABSTRACT

Panoptic Video Scene Graph Generation (PVSG) aims to achieve a comprehensive video understanding by segmenting entities and predicting their temporal relations. These temporal relations vary in duration and evolve dynamically over time. However, existing methods model relations over the entire video sequence, making it difficult to align the perception scope with actual interaction intervals and often introducing irrelevant context. To address this, we propose **T**emp**F**ocus**Net** (**TFNet**), a new framework that first localizes the intervals where relations occur and then performs focused context modeling within them, enabling temporally aligned and more accurate relation prediction. Specifically, we extract visual and category semantic features for each entity to construct temporally continuous entity feature tubes. Then, multiple temporal queries interact with paired entity tubes to capture diverse temporal cues and generate candidate relation intervals, which are represented as Gaussian masks to model their temporal structure. Finally, the Gaussian masks guide the temporal focus attention to focus on relevant intervals for relation classification. Extensive experiments show that our TFNet achieves state-of-the-art performance on OpenPVSG and ImageNet-VidVRD datasets. The code of our TFNet will be made available.

## 1 INTRODUCTION

Understanding complex visual scenes in the video requires not only identifying objects, but also modeling their fine-grained interactions and temporal dynamics. Panoptic Video Scene Graph Generation (PVSG) aims to capture such complex visual semantics by constructing a dynamic scene graph, where each node represents a pixel-level entity, and each edge encodes the interaction between nodes within specific temporal intervals. This structured representation of videos facilitates advancement in a wide range of downstream tasks, including visual reasoning (Chu et al., 2025; Qiu et al., 2025), video question answering (Li et al., 2022b; Min et al., 2024), video captioning (Tian et al., 2025; Yang et al., 2023a), and dynamic environment guidance (Lan et al., 2025; Ye et al., 2023).

In the PVSG task, relations exhibit significant temporal complexity, with diverse durations and dynamic evolution. These temporal characteristics require flexible and adaptive modeling strategies. However, existing methods (Yang et al., 2023c;b; Nguyen et al., 2025a) model relations over the entire video span, taking full temporal trajectories of entities into account to jointly localize and classify relations. This strategy struggles to capture the sparse and diverse nature of PVSG relations, which inevitably introduce irrelevant or noisy frames, leading to fragmented or incorrect classification of relations, as shown in Figure 1(a).

To address the critical issues of relation diversity in PVSG, we argue that it is essential to decouple the task of localizing a relation interval from classifying its category. Therefore, we introduce a localization-then-classification framework designed specifically for this purpose. We first use a global view to localize the temporal intervals where relations occur, and then guide relation classification to operate within them, as illustrated in Figure 1(b). This approach significantly reduces the impact of irrelevant context, improves prediction accuracy and robustness, and presents a new perspective for modeling temporal dynamics in PVSG.

Although some work on similar tasks such as Video Visual Relation Detection(VidVRD) (Liu et al., 2020; Chen et al., 2021) attempt to locate interaction intervals via sliding windows or frame-wise

Figure 1: Previous methods predict relations over the entire video, overlooking their temporal locality. In contrast, our TFNet first locates relation intervals and then performs fine-grained classification within each interval.

scoring, but they suffer from a limited receptive field and lack holistic modeling over the entire relation interval. As a result, they struggle to capture the temporal dynamics of relations in PVSG. Motivated by the success of query-based object localization(Carion et al., 2020; Cheng et al., 2022), we introduce a novel approach that employs temporal queries to model global relation dynamics and directly regress relation intervals in terms of their center and width. These queries are encouraged to capture relation intervals with diverse temporal characteristics, such as varying durations or dynamic patterns. Moreover, by modeling intervals through center-width regression, the model learns patterns and duration distributions in a holistic manner. Overall, this formulation enables flexible and context-aware temporal reasoning, allowing the model to handle the diverse and evolving nature of relations effectively.

With the above insight, we propose a new localization-then-classification framework for the PVSG task, termed TempFocusNet(TFNet). By decoupling relation interval localization from relation classification and leveraging the above query-based design for temporal grounding, the framework achieves both accurate temporal localization and robust relation prediction. Specifically, our model comprises three parts. First, the *Entity Feature Extractor* extracts visual and category semantic features for each entity, and then fuses them to capture both interaction cues and relation priors. The fused features are then tracked and paired to form final entity feature pairs. Next, we introduce *Relation Interval Locator*, which employs multiple temporal queries to flexibly locate candidate relation intervals across the video. These queries attend to entity feature pairs through a Temporal Query Decoder and directly regress the centers and widths of relation intervals. A Gaussian mask is then generated for each interval to represent its temporal structure. Finally, the *Temporal Focus Classifier* leverages Gaussian masks to modulate attention weights within Temporal Focus Attention, guiding the model to focus on the most relevant temporal segments for accurate relation classification. Through the above design, our model enables precise temporal relation modeling. Meanwhile, by decoupling relation span localization from classification, our framework supports flexible component replacement and iterative refinement, laying a solid foundation for advancing video understanding.

To sum up, our contributions are as follows: **1)** We propose TempFocusNet (TFNet), a localization-then-classification framework that enables precise modeling of dynamic temporal relations in PVSG. **2)** We introduce a query-based Relation Interval Locator that predicts interaction spans with diverse temporal characteristics in a holistic and context-aware manner. **3)** Experiments on OpenPVSG and ImageNet-VidVRD demonstrate our method significantly outperforms existing PVSG methods.

## 2 RELATED WORK

### 2.1 PANOPTIC VIDEO SCENE GRAPH GENERATION

Scene Graph Generation (SGG) aims to detect objects and infer their semantic relations within a single image, yielding a structured representation of visual content (Yang et al., 2022; Zhou et al., 2024; Fu et al., 2025). To extend visual relation reasoning into the temporal domain, Video Scene Graph Generation (VidSGG) models spatial-temporal interactions across consecutive frames (Peddi et al., 2025; Nguyen et al., 2025b; Li et al., 2025).

Panoptic Video Scene Graph Generation (PVSG) is a recent extension of VidSGG; both tasks seek to reason about evolving interactions over time (Yang et al., 2023b;c). However, PVSG replaces coarse bounding boxes with pixel-level panoptic masks that represent object instances and amorphous backgrounds. This precise representation offers finer detail, clearer semantics, and continuous spatial cues that enhance relation reasoning under motion and occlusion. While existing PVSG studies mainly emphasize motion cues or cross-dataset adaptation (Nguyen et al., 2025a; Wu et al., 2025), they overlook the gap between relational dynamics and temporal modeling. To bridge this gap, we first localize interaction intervals, constraining relation modeling to appropriate temporal scales and achieving more accurate predictions of dynamic relations.

## 2.2 TEMPORAL RELATION INTERVAL LOCALIZATION

To accurately capture the temporal intervals of relational events, early studies often divided videos into fixed-length segments (Shang et al., 2017; Su et al., 2020; Wei et al., 2024), predicted the relations within them, and concatenated the results along the timeline. While these segment-based strategies are straightforward, they rely heavily on heuristic post-processing, making them susceptible to boundary drift and missed detections. Later work shifted to trajectory-based relation modeling. Some methods (Chen et al., 2021) assign frame-level interaction scores to pairs of object-trajectory and cluster high-scoring intervals, while others (Woo et al., 2021; Jiang et al., 2024; Gao et al., 2022) define temporal anchor segments along trajectories for binary classification or boundary regression. Although these approaches mitigate fragmentation to some degree, they still generate redundant candidates and struggle to balance short-term dynamics with long-range dependencies.

Temporal modeling in PVSG is more challenging: relations may evolve gradually or shift abruptly, occurring between both objects and large-scale background regions. These relations can recur across disjoint intervals and frequently overlap with other relations in time. Most existing PVSG methods rely on frame-level relation existence prediction (Yang et al., 2023c; Nguyen et al., 2025a), which struggles to capture such complex temporal patterns. To tackle this challenge, we introduce a query-based interval locator that directly predicts relation centers and widths from global context, avoiding handcrafted windows and enabling flexible modeling of complex temporal dynamics.

## 3 TASK DEFINITION

Given a video, the PVSG task aims to construct a dynamic scene graph where entities are represented as nodes and their spatio-temporal relation are encoded as edges. Formally, the input of the PVSG model is a video clip $V \in \mathbb{R}^{T \times H \times W \times 3}$, where $T$ denotes the number of frames, and the frame size $H \times W$ is consistent across the video. The output of the model is a dynamic scene graph $G$. The PVSG task can be formulated as follows:

$$P(G|V) = P(M, O, R|V). \tag{1}$$

Specifically, the scene graph $G$ consists of a set of binary mask tubes $M = \{m_1, \ldots, m_N\}$, entity category labels $O = \{o_1, \ldots, o_N\} \in \mathbb{C}^O$, and temporal relations $R = \{r_1, \ldots, r_L\} \in \mathbb{C}^R$. Each entity $i$ is represented by a spatio-temporal mask tube $m_i \in \{0, 1\}^{T \times H \times W}$ and a category label $o_i \in \mathbb{C}^O$. Each relation $r_l \in \mathbb{C}^R$ encodes an interaction between a subject-object pair, annotated with a relation type and a time period.

## 4 METHODOLOGY

### 4.1 OVERVIEW

As illustrated in Figure 2, our TempFocusNet (TFNet) comprises three components: *Entity Feature Extractor*, *Relation Interval Locator*, and *Temporal Focus Classifier*. For the *Entity Feature Extractor*, we use a segmentation model to extract entity features consisting of both visual information and categorical semantics, then track and selectively pair them to form entity feature pairs. The *Relation Interval Locator* leverages multiple temporal queries to predict their diverse temporal relation intervals, each modeled by a corresponding Gaussian mask to capture its temporal structure. Finally, the *Temporal Focus Classifier* utilizes Gaussian masks to guide Temporal Focus Attention to focus on each predicted interval and assigns corresponding relation scores. A score-based merging strategy then merges intervals with high confidence to generate the final scene graph.

Figure 2: The pipeline of our method. Given a video, we first apply Mask2Former (Cheng et al., 2022) to extract visual features $f_v$ and category semantic features $f_s$ for each entity. These features are fused, tracked across time, and selectively paired to form entity feature pairs $F_{ij}$. For each pair, a set of temporal queries $Q$ is used to predict multiple relation intervals, each with a center $c_k$, width $w_k$, and a corresponding Gaussian mask $g_k$ that models its temporal structure. Then, these masks guide the Temporal Focus Attention to concentrate on each interval and predict the relation score $\hat{y}$. Finally, a score-based merging strategy merges intervals with high confidence to produce the final relation predictions.

## 4.2 ENTITY FEATURE EXTRACTOR

The Entity Feature Extractor aims to obtain temporally consistent representations for each entity by extracting per-frame entity features and associating them across time. Given a frame in video $V$, we apply Mask2Former (Cheng et al., 2022) to extract a feature map through its backbone and produce a set of masks with corresponding categories. For each detected entity, we perform mask pooling over the feature map using its corresponding mask to obtain the visual representation $f_v \in \mathbb{R}^D$, which encodes critical cues for localizing interaction timings. In parallel, the predicted category index is mapped to a learnable semantic embedding $f_s \in \mathbb{R}^D$, offering strong priors for relation understanding. We concatenate $f_v$ and $f_s$, and pass them through an MLP to construct the final entity feature $f_e$ as follows:

$$f_e = f_v + \alpha \cdot \text{MLP}([f_v, f_s]), \tag{2}$$

where $f_e \in \mathbb{R}^D$, $D$ denote the dimension of feature, and $\alpha$ is a learnable gating coefficient.

Afterwards, we associate entities across the temporal dimension and concatenate the features corresponding to the same entity across different timestamps to form entity feature tubes. We consider two approaches: 1) Image Panoptic Segmentation combined with a Tracker (IPS+T), where a tracker (Wang et al., 2021) links mask tubes across video frames, and 2) Video Panoptic Segmentation (VPS), where each frame is processed in conjunction with a nearby reference frame to maintain temporal consistency (Li et al., 2022a). Both approaches produce a set of entity feature tubes $\{F_i\}_{i=1}^N$, where $F_i \in \mathbb{R}^{T \times D}$.

Next, we construct entity feature pair $F_{ij} = [F_i, F_j] \in \mathbb{R}^{T \times 2D}$ by concatenate the features tubes of entities $i$ and $j$. Since not all pairs are relevant for relation modeling, only pairs matched with ground-truth are used for supervision. During inference, we employ the pairing component (Yang et al., 2023c; Wang et al., 2024) to efficiently select candidate entity pairs based on pairwise similarity.

## 4.3 RELATION INTERVAL LOCATOR

To localize and model dynamic relations, we first employ a Temporal Query Decoder to generate diverse candidate relation intervals. These intervals are then encoded as Gaussian masks to model fine-grained temporal dynamics.

**Temporal Query Decoder.** Given a selected entity pair $F_{ij} \in \mathbb{R}^{T \times 2D}$, we denote it as $F$ for brevity. A linear layer is first applied to reduce the dimension to $D$, followed by a self-attention layer $SA(\cdot)$ to encode the temporal context of the interaction and produce a refined representation $\bar{F} \in \mathbb{R}^{T \times D}$. Then, $K$ temporal queries $Q \in \mathbb{R}^{K \times D}$ attend to the encoded sequence $\bar{F}$ via a cross-attention layer $CA(\cdot)$ and produce a set of interaction features $H \in \mathbb{R}^{K \times D}$. This design enables each learnable query to capture global interaction context and adaptively localize diverse and dynamic relation intervals in PVSG. The operation is formulated as

$$\bar{F} = SA\left(\text{Linear}(F)\right), \quad H = CA\left(Q, \bar{F}, \bar{F}\right). \tag{3}$$

Since the interaction features $H$ aggregate temporal information across time, we use them to predict relation intervals by applying a linear layer and a sigmoid activation to obtain a normalized 2D vector.

$$I = \text{Sigmoid}\left(\text{Linear}\left(H\right)\right) \in [0, 1]^{K \times 2}, \tag{4}$$

where $I = \{I_k\}_{k=1}^{K}$ and each $I_k = (c_k, w_k)$ denotes the normalized predicted center and width of the $k$-th interval. The final temporal interval $\hat{I}_k$ is calculated as the range $[c_k - w_k/2, c_k + w_k/2]$.

To associate each predicted interaction interval $\hat{I}_k$ with a relevant ground-truth annotation $\hat{I}_k^{gt}$, we calculate the temporal IoU between $\hat{I}_k$ and all ground-truth intervals, and select the one with the highest IoU as $\hat{I}_k^{gt}$. If all IoU values are zero, we instead match $\hat{I}_k$ to the ground-truth with the smallest temporal endpoint distance, defined as the sum of absolute differences between their start and end timestamps. The matching process is formally defined as:

$$\hat{I}_k^{gt} = \begin{cases} \arg\max_j \text{IoU}\left(\hat{I}_k, \hat{I}_j^{gt}\right), & \text{if } \max_j \text{IoU}\left(\hat{I}_k, \hat{I}_j^{gt}\right) > 0 \\ \arg\min_j \text{Dist}\left(\hat{I}_k, \hat{I}_j^{gt}\right), & \text{otherwise}, \end{cases} \tag{5}$$

where $\text{Dist}(\cdot)$ denotes the temporal endpoint distance. To train the locator, we define the localization loss as:

$$\mathcal{L}_{\text{loc}} = -\frac{1}{K} \sum_{k=1}^{K} \log\left(\text{IoU}(\hat{I}_k, \hat{I}_k^{gt})\right). \tag{6}$$

**Gaussian mask.** To improve the modeling of relation intervals and provide focused guidance for subsequent temporal focus classification, we adopt Gaussian masks to represent interval attention. Inspired by CNM (Zheng et al., 2022), these masks are used to capture the inherent temporal structure of relations (including start, peak, and end). Specifically, the Gaussian masks $g \in \mathbb{R}^{K \times T}$ are formulated as:

$$g_k^t = \frac{1}{\sqrt{2\pi}\left(w_k/\tau\right)} exp\left(\frac{-\left(t/T - c_k\right)^2}{2\left(w_k/\tau\right)^2}\right), \tag{7}$$

where $t = 1, \ldots, T$ is the frame index, $g_t^k$ is the weight of the $t$-th frame in the $k$-th Gaussian mask, and $\tau$ is a hyperparameter that controls the variance of the Gaussian function. Additionally, we generate Gaussian masks $g_k^{gt}$ for the corresponding ground truth interval to provide more fine-grained structural constraints for the predicted intervals.

$$\mathcal{L}_{\text{mask}} = \frac{1}{K} \sum_{k=1}^{K} \text{MSE}\left(g_k, g_k^{gt}\right). \tag{8}$$

Finally, we apply a diversity loss (Lin et al., 2017) to encourage the predicted interval masks to capture a variety of temporal patterns. This regularization forces different queries to attend to relation intervals with diverse temporal characteristics and durations.

$$\mathcal{L}_{\text{div}} = \left|\left|gg^T - \lambda\mathbb{I}\right|\right|_F^2, \tag{9}$$

where $||\cdot||_F$ stands for the Frobenius norm of a matrix, and $\lambda$ is a hyperparameter that controls the degree of diversity among the predicted masks.

## 4.4 TEMPORAL FOCUS CLASSIFIER

After localizing the interaction intervals, we introduce a Temporal Focus Classifier to predict the relation category for each interval. To this end, a mask-guided attention mechanism, called *Temporal Focus Attention*, is employed to enhance relation classification with temporal awareness. Specifically, we use Gaussian masks $g$ from the predicted intervals to modulate the attention weights and guide the model to focus on the most relevant temporal regions when a relation occurs. The mechanism operates by first applying a linear layer to project entity pair feature $F \in \mathbb{R}^{T \times 2D}$ to the attention query $Q_a \in \mathbb{R}^{T \times D}$, keys $K_a \in \mathbb{R}^{T \times D}$ and values $V_a \in \mathbb{R}^{T \times D}$. Then the attention map is calculated as $\mathbb{A} = \frac{Q_a K_a^T}{\sqrt{D}}$. To guide the attention toward the predicted intervals, we apply the corresponding Gaussian mask $g_k \in \mathbb{R}^T$ to the attention map via row-wise multiplication. The final focusing attention feature for the $k$-th interval is given by:

$$\hat{F}_k = \text{Softmax}\left(\mathbb{A} \otimes g_k\right) \cdot V_a \in \mathbb{R}^{T \times D}, \tag{10}$$

where $\otimes$ denotes the row-wise multiplication. Next, an MLP transforms $\hat{F}_k$ into a score matrix of shape $\mathbb{R}^{T \times C}$, where $C = |\mathbb{C}^R|$ denotes the number of relation categories. Max pooling is then applied along the time dimension to compress the sequence of scores into a single $C$-dimensional vector:

$$\hat{y}_k = \text{Pooling}\left(\text{MLP}\left(\hat{F}_k\right)\right) \in \mathbb{R}^C. \tag{11}$$

Since multiple relations may occur simultaneously for the same entity pair, predicting the relation category for a single interval is formulated as a multi-label classification task. For each interval, we compute the temporal IoU with all ground-truth relation intervals. Categories corresponding to ground-truth intervals with IoU greater than 0.5 are selected as positive labels, forming a binary multi-hot supervision vector $y$. The classification loss is computed using binary cross-entropy:

$$\mathcal{L}_{\text{cls}} = \frac{1}{K} \sum_{k=1}^{K} \text{BCE}\left(\hat{y}_k, y_k\right), \tag{12}$$

During training, predicted intervals may fail to align with ground-truth, leading to insufficient supervision. To address this, we use ground-truth intervals and their corresponding Gaussian masks to train the Temporal Focus Classifier.

## 4.5 MODEL TRAINING AND INFERENCE

In this section, we describe the loss function to optimize our TFNet and the inference process.

### 4.5.1 TRAINING.

We train our model with four loss terms: 1) localization loss $\mathcal{L}_{\text{loc}}$, 2) mask alignment loss $\mathcal{L}_{\text{mask}}$, 3) diversity loss $\mathcal{L}_{\text{div}}$, and 4) classification loss $\mathcal{L}_{\text{cls}}$. The overall training objective is formulated as:

$$\mathcal{L}_{\text{total}} = \mathcal{L}_{\text{loc}} + \alpha_1 \mathcal{L}_{\text{mask}} + \alpha_2 \mathcal{L}_{\text{div}} + \alpha_3 \mathcal{L}_{\text{cls}}, \tag{13}$$

where $\alpha_1$, $\alpha_2$ and $\alpha_3$ are hyperparameters to balance losses.

### 4.5.2 INFERENCE.

For each entity pair, the model predicts $K$ independent relation intervals along with their corresponding relation scores $\hat{y} \in \mathbb{R}^{K \times C}$. To produce final relation instances in the same format as ground-truth for evaluation, we apply a Score-based Merging Strategy to merge intervals based on their relation scores. For a given relation category $c$, there are K relation scores $y_k^c$. We define a threshold to select the predicted intervals for the $c$:

$$\hat{I}_c^{\text{select}} = \left\{ \hat{I}_k \,\middle|\, \hat{y}_k^c \geq \hat{y}_{max}^c - \delta \right\}, \tag{14}$$

where $\delta$ is the margin and $\hat{y}_{max}^c = \max_{k=1}^{K}(\hat{y}_k^c)$. Then we merge $\hat{I}_c^{\text{select}}$ to represent the temporal span of $c$, and the final relation score for retrieval is formulated as:

$$\hat{y}^c = \text{mean}\left(\left\{ \hat{y}_k^c \,\middle|\, \hat{I}_k \in \hat{I}_c^{\text{select}} \right\}\right). \tag{15}$$

Table 1: Comparison between TFNet and other methods on the OpenPVSG dataset in both IPS+T and VPS settings.

| Method | | threshold = 0.5 | | | threshold = 0.1 | | |
|---|---|---|---|---|---|---|---|
| | | R/mR@20 | R/mR@50 | R/mR@100 | R/mR@20 | R/mR@50 | R/mR@100 |
| IPS+T | Vanilla | 3.04 / 1.35 | 4.61 / 2.94 | 5.56 / 3.33 | 8.28 / 5.68 | 14.47 / 9.92 | 18.24 / 11.84 |
| | Handcrafted filter | 2.52 / 1.72 | 3.77 / 2.36 | 4.72 / 2.79 | 8.07 / 5.61 | 13.42 / 8.27 | 16.46 / 10.11 |
| | Transformer | 3.88 / 2.81 | 5.66 / 4.12 | 6.18 / 4.44 | 9.01 / 6.69 | 14.88 / 11.28 | 17.51 / 13.20 |
| | Convolution | 3.88 / 2.55 | 5.24 / 3.29 | 6.71 / 5.36 | 10.06 / 8.98 | 14.99 / 12.21 | 18.13 / 15.47 |
| | Motion-aware Transformer | 3.98 / 2.98 | 5.97 / 4.20 | 7.44 / 5.15 | 10.59 / 9.56 | 16.98 / 12.39 | 22.33 / 17.47 |
| | Motion-aware Convolution | 4.51 / 3.56 | 6.08 / 4.38 | 7.76 / 5.86 | 11.43 / 9.57 | 17.30 / 13.13 | 22.85 / 17.48 |
| | **TFNet (Ours)** | **5.35 / 6.02** | **10.48 / 10.12** | **12.89 / 12.87** | **13.84 / 13.50** | **23.27 / 21.34** | **28.93 / 26.50** |
| VPS | Vanilla | 0.21 / 0.10 | 0.21 / 0.10 | 0.31 / 0.18 | 6.29 / 3.04 | 9.64 / 6.74 | 12.89 / 9.60 |
| | Handcrafted filter | 0.42 / 0.13 | 0.52 / 0.50 | 0.94 / 0.92 | 5.24 / 2.84 | 7.65 / 7.14 | 9.64 / 8.22 |
| | Transformer | 0.42 / 0.61 | 0.73 / 0.76 | 1.05 / 0.92 | 6.50 / 5.75 | 9.64 / 8.25 | 12.26 / 9.51 |
| | Convolution | 0.42 / 0.25 | 0.63 / 0.67 | 0.63 / 0.67 | 8.07 / 7.84 | 11.01 / 9.78 | 12.89 / 10.77 |
| | Motion-aware Transformer | 0.63 / 0.83 | 1.05 / 0.76 | 1.05 / 0.76 | 6.71 / 6.94 | 10.27 / 8.68 | 13.42 / 12.09 |
| | Motion-aware Convolution | 0.84 / 0.98 | 1.26 / 1.22 | 1.26 / 1.22 | 8.18 / 8.00 | 12.90 / 11.47 | 14.22 / 13.59 |
| | **TFNet (Ours)** | **1.78 / 1.52** | **2.52 / 2.16** | **3.04 / 2.79** | **8.39 / 8.45** | **15.72 / 15.33** | **22.01 / 21.63** |

## 5 EXPERIMENTS

### 5.1 EXPERIMENT SETTING

**Dataset.** We evaluate our method on OpenPVSG (Yang et al., 2023c), a benchmark for PVSG tasks, containing 400 video clips (289 third-person and 111 egocentric), with an average duration of 77 seconds. The dataset includes 153K annotated frames, 7.6K object mask tracks (126 categories), and 4548 relation triplets (57 relation types). We follow the standard split from prior work (Yang et al., 2023c; Nguyen et al., 2025a), with 338 training and 62 testing videos. Due to the scarcity of PVSG datasets, we additionally adopt ImageNet-VidVRD (Shang et al., 2017), a VidSGG dataset containing 1000 videos (800 for training and 200 for testing). It includes 35 object categories and 132 relation types.

**Evaluation Metrics.** Following prior works (Yang et al., 2023c; Nguyen et al., 2025a), we employ recall at K (R@K) and mean recall at K (mR@K) to evaluate the top-K relation triplets predicted by PVSG models. R@K measures overall retrieval performance across all relations, while mR@K computes the relation-averaged recall, highlighting performance on rare or long-tail relations. For a predicted triplet to be counted as correctly recalled, both of the following conditions should be satisfied: (1) the correct category labels of the subject, object, and relation; (2) the volume-IoU (vIoU) between the predicted mask tubelet and ground-truth mask tubelet reaching a specified threshold of 0.5. To ensure comprehensive and fair comparison across methods, we additionally report results under a relaxed setting with a weak vIoU threshold of 0.1.

**Implementation Details.** Following prior works (Yang et al., 2023c; Nguyen et al., 2025a), we employ the identical IPS+T and VPS as the segmentation module in all experiments for fair comparisons. For IPS+T, a fine-tuned Mask2Former (Cheng et al., 2022) performs frame-level panoptic segmentation, and a pretrained UniTrack (Wang et al., 2021) constructs object tubes. For VPS, we adopt Video K-Net (Li et al., 2022a) integrated within the Mask2Former framework to achieve video panoptic segmentation. Both segmentation models are built upon Mask2Former with a ResNet-50 (He et al., 2016) backbone, trained for 8 epochs, kept frozen during TFNet training. We optimize our model using the AdamW (Loshchilov & Hutter, 2017) optimizer with a learning rate of $1e^{-4}$ and a weight decay of $1e^{-2}$. The model is trained for 100 epochs on an RTX 4090. The feature dimension is fixed at $D = 256$. For the hyperparameters, we set $K = 8$, $\tau = 9$, $\lambda = 0.15$, $\alpha_1 = 0.5$, $\alpha_2 = 2.5$, $\alpha_3 = 0.5$. On ImageNet-VidVRD, we adopt the object tubes extracted by MEGA (Chen et al., 2020) and set $D = 512$, with a learning rate of $5e^{-5}$, keeping other settings consistent with OpenPVSG.

### 5.2 COMPARISON WITH STATE-OF-THE-ART METHODS

**Compared Methods.** In both the IPS+T and VPS settings, we compare our method with the baseline approaches PVSG (Yang et al., 2023c) and Motion-aware (Nguyen et al., 2025a). From PVSG, we consider: (1) Vanilla, applying fully-connected layers to separately encode temporal states of entity mask tube; (2) Handcrafted Filter, using a fixed kernel to aggregate contextual information from

adjacent frames; (3) Transformer Encoder, encoding entity feature tubes with a transformer encoder block; (4) 1D Convolution, employing a learnable one-dimensional convolution to capture temporal information. From Motion-aware, we adopt: (5) Motion-Aware Transformer, which adds a motion-aware contrastive learning framework to the Transformer baseline; (6) Motion-Aware Convolution, which applies the same motion-aware contrastive learning framework to the 1D Convolution baseline. These baselines together represent the current state-of-the-art methods in PVSG.

**Main Results.** Table 1 provides the main experimental results on OpenPVSG. TFNet consistently outperforms all existing methods. With a vIoU threshold of 0.5 in the IPS+T setting, TFNet raises R/mR by +0.84/+2.46 at @20, +4.40/+5.74 at @50, and +5.13/+7.01 at @100 over the previous best. For the vIoU threshold of 0.1, TFNet still improves R/mR@20 by +2.41/+3.93 over the Motion-Aware Convolution baseline. Notably, the improvement in mRecall is more significant, suggesting that TFNet achieves a more balanced prediction across relation categories. In the VPS setting, where spatio-temporal masks are less accurate, TFNet likewise surpasses all competitors, further confirming its robustness. These results demonstrate the effectiveness of temporally focused modeling in PVSG.

**Evaluation across Relation Categories.** We report Normalized Recall@20 for each relation category, defined as the ratio of obtained Recall@20 to the maximum achievable recall under the IPS+T setting. Only categories with at least one correct prediction by either method are shown. As shown in Figure 3, our method consistently outperforms the Transformer baseline across most categories, with notably greater improvements in rare relations.

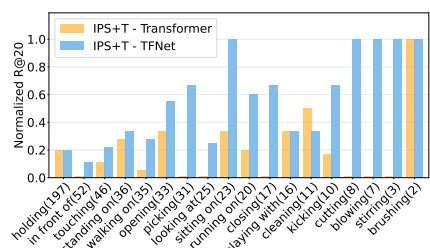

Figure 3: Normalized R@20 across relation categories for Transformer and TFNet. Numbers in parentheses are sample counts.

**Evaluation on ImageNet-VidVRD.** We conduct additional experiments on ImageNet-VidVRD, which provides only bounding box annotations. Accordingly, we compute vIoU for bbox tubelets instead of masks. All other evaluation settings remain consistent with previous experiments, with a vIoU threshold of 0.5. We compare against both PVSG baselines and state-of-the-art VidVRD methods, including BIG-C and VrdONE (Gao et al., 2022; Jiang et al., 2024). As shown in Table 2, our method achieves outstanding performance, validating the generalization capability of our framework.

## 5.3 ABLATION STUDY

We perform ablation studies on OpenPVSG under the IPS+T setting to evaluate each module in TFNet. We provide more analyses in the Appendix, including loss and hyperparameter ablations as well as more visualization results.

**Comparison of Temporal Modeling Strategies.** To evaluate the impact of different temporal modeling strategies on relation classification, we replace the Temporal Focus Attention module with a convolution layer or a transformer encoder. As shown in Table 3, TFNet outperforms both alternatives. This improvement is due to the use of Gaussian masks, which guide the model to focus on the relation intervals where relations occur, highlighting the advantage of relation modeling at an appropriate temporal scale.

**Comparison of Relation Temporal Localization Capability.** To assess the Relation Interval Locator, we retrieve the top 5000 relation predictions. These operation enumerates all relation categories, minimizing the influence of relation classification on recall. We report R@5000, wR@5000, and soft R@5000 for TFNet and PVSG baselines (Yang et al., 2023c). Among these metrics, soft recall accumulates actual vIoU scores above a

Table 2: Performance on the ImageNet-VidVRD dataset.

| Method | R/mR@20 | R/mR@50 | R/mR@100 |
|---|---|---|---|
| Transformer | 8.19 / 5.18 | 11.71 / 8.19 | 14.50 / 11.14 |
| Convolution | 11.79 / 7.80 | 17.61 / 11.70 | 22.24 / 15.02 |
| BIG-C | 13.00 / 9.46 | 18.13 / 14.32 | 24.27 / 18.98 |
| VrdONE | **13.96** / 8.51 | 18.62 / 11.52 | 21.58 / 13.03 |
| **TFNet (Ours)** | 13.09 / **10.01** | **21.71 / 17.64** | **29.18 / 22.66** |

Table 3: Ablation study comparing different temporal modeling strategies.

| Backbone | R/mR@20 | R/mR@50 | R/mR@100 |
|---|---|---|---|
| Convolution | 4.61 / 5.05 | 7.02 / 7.42 | 9.96 / 11.57 |
| Transformer | 3.35 / 4.14 | 7.23 / 9.28 | 11.53 / 12.23 |
| **Ours** | **5.35 / 6.02** | **10.48 / 10.12** | **12.89 / 12.87** |

Table 4: Ablation study on relation temporal localization capability across methods.

| Method | R@5000 | w R@5000 | soft R@5000 |
|---|---|---|---|
| Vanilla | 6.60 | 22.75 | 4.81 |
| Handcrafted filter | 7.34 | 25.15 | 5.16 |
| Transformer | 9.01 | 27.57 | 6.57 |
| Convolution | 10.38 | 27.78 | 7.43 |
| **TFNet (Ours)** | **21.49** | **45.91** | **16.67** |

Figure 4: Qualitative comparison between our method and IPS+T-Transformer (Yang et al., 2023c).

threshold, offering a finer measure of localization quality. As shown in Table 4, TFNet consistently outperforms all baselines, demonstrating that the proposed query-based localization framework enables more flexible modeling of the complex temporal dynamics and thereby achieves superior relation interval localization performance.

**Effect of Entity Feature Representations.** To assess the effect of different entity feature representations, we perform three ablation settings: (1) directly using the query in Mask2Former as the entity feature (Yang et al., 2023c), (2) using only the visual features obtained via mask pooling, and (3) our full method, which augments the visual features with category semantic embeddings. As reported in Table 5, visual features lead to clear improvements over query features, as they provide essential cues for interaction interval localization. Furthermore, combining visual features with category semantic embedding brings additional improvements, since entity categories offer strong priors for relation prediction.

Table 5: Ablation study comparing different entity features.

| Entity Feature | R/mR@20 | R/mR@50 | R/mR@100 |
|---|---|---|---|
| Query feature | 4.51 / 5.51 | 7.86 / 7.33 | 12.16 / 10.74 |
| Visual feature | 4.93 / 5.94 | 8.91 / 9.13 | 11.84 / 10.98 |
| **Vis + Semantic** | **5.35 / 6.02** | **10.48 / 10.12** | **12.89 / 12.87** |

## 5.4 QUALITATIVE ANALYSIS

**Visualization Results.** In Figure 4, we present some visual results with a comparison to our TFNet and IPS+T-Transformer (Yang et al. 2023) on OpenPVSG. In Figure 4(a), TFNet successfully identifies all three relations and correctly localizes their respective intervals. In contrast, the baseline fails to retrieve *chasing* and *kicking*, highlighting its limited capacity to capture complex temporal dynamics relations. In Figure 4(b), TFNet provides cleaner and more aligned intervals for *standing on* and *walking on*, avoiding fragmented predictions. This example demonstrates that our method effectively captures subtle and transient interactions. Across all examples, TFNet demonstrates superior performance in both relation classification and temporal localization.

**Analysis of Failure Cases.** Despite the overall effectiveness of TFNet, some failure cases reveal challenges in the upstream perception modules. As shown in Figure 4(a), the tracker fails to maintain consistent instance IDs for the same subject (child) and object (ball) across frames. Consequently, TFNet splits the relation *playing with* into two separate segments. In Figure 4(b), segmentation errors lead to missing masks for the object (floor) in the middle of the relation, resulting in a fragmented prediction for the *standing on* relation. These cases highlight the dependency of relation reasoning on robust entity tracking and segmentation, suggesting potential benefits from tighter integration with these components.

## 6 CONCLUSION

In this paper, we propose TempFocusNet(TFNet), a new framework for panoptic video scene graph generation that enables temporally aligned relation modeling. Unlike previous approaches that model dynamic relations over the entire video sequence, TFNet first localizes the temporal intervals where relations occur and then performs accurate relation prediction within these intervals. To this end, we introduce a new interval localization module that generates candidate intervals using multiple temporal queries and encodes their temporal structure via Gaussian masks. Furthermore, these masks guide a temporal focus attention mechanism to enhance relation prediction within the most relevant moments. Extensive experiments demonstrate that our method achieves excellent performance in panoptic video scene graph generation.

## 7 ETHICS STATEMENT

This work adheres to the ICLR Code of Ethics. In this study, no human subjects or animal experimentation were involved. All datasets used, including OpenPVSG and ImageNet-VidVRD, were sourced in compliance with relevant usage guidelines, ensuring no violation of privacy. We have taken care to avoid any biases or discriminatory outcomes in our research process. No personally identifiable information was used, and no experiments were conducted that could raise privacy or security concerns. We are committed to maintaining transparency and integrity throughout the research process.

## 8 REPRODUCIBILITY STATEMENT

We have made every effort to ensure that the results presented in this paper are reproducible. All code and datasets have been made publicly available in an anonymous repository to facilitate replication and verification. The experimental setup, including training steps, model configurations, and hardware details, is described in detail in the paper. We have also provided a full description of our proposed TempFocusNet (TFNet) framework and its implementation details to assist others in reproducing our experiments. Additionally, publicly available video scene graph benchmarks, such as OpenPVSG and ImageNet-VidVRD, are publicly available, ensuring consistent and reproducible evaluation results. We believe these measures will enable other researchers to reproduce our work and further advance the field.

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

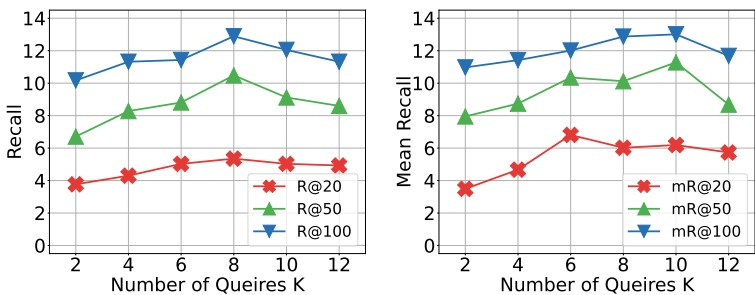

Figure 5: Ablation results on different numbers of queries.

Table 6: Ablations of each loss item for TFNet.

| $\mathcal{L}_{\text{loc}}$ | $\mathcal{L}_{\text{mask}}$ | $\mathcal{L}_{\text{div}}$ | R/mR@20 | R/mR@50 | R/mR@100 |
|---|---|---|---|---|---|
| | ✓ | ✓ | 1.36 / 1.53 | 3.25 / 2.52 | 3.98 / 3.51 |
| ✓ | | ✓ | 4.72 / 5.76 | 8.70 / 8.11 | 12.05 / 10.62 |
| ✓ | ✓ | | 4.61 / 5.20 | 8.91 / 8.60 | 12.05 / 10.44 |
| ✓ | ✓ | ✓ | **5.35 / 6.02** | **10.48 / 10.12** | **12.89 / 12.87** |

# A APPENDIX

In this supplementary material, we present additional details and results that could not be included in the main paper due to space limitations. First, we conduct further ablation studies, including analyses of hyperparameter settings and loss components. Then, we provide additional qualitative visualizations.

## A.1 MORE ABLATION STUDY

We present the ablation experiment results on the OpenPVSG datasets with the setting of IPS+T as the segmentation module. We report Recall at K (R@K) and mean Recall at K (mR@K) metrics under the vIoU threshold of 0.5. The experimental settings remained consistent with those presented in the main paper.

### A.1.1 EFFECT OF THE NUMBER OF TEMPORAL QUERIES.

To examine the impact of the number of learnable temporal queries, we vary $K$ over values $\{2, 4, 6, 8, 10, 12\}$ and compare their performance. As shown in Figure 5, a moderate number of temporal queries yields the best results. $K = 8$ gives the highest recall and $K = 10$ provides the best mean recall. When $K$ is too small, the queries cannot cover all potential relation intervals, leading to missed detections. Conversely, a large $K$ exceeds the actual demand of the task, causing conflicts between localization and diversity, which leads to unstable training. Considering the trade-off between accuracy and efficiency, we choose $K = 8$ as a practical setting.

### A.1.2 IMPACT OF LOSSES FOR RELATION INTERVAL LOCATOR.

We conducted an ablation study to examine the importance of each loss item in the Relation Interval Locator. The results in Table 6 reveal that $\mathcal{L}_{\text{loc}}$ is the most crucial loss item under this setting, as it directly constrains the accuracy of predicted relation intervals, ensuring reliable temporal grounding for downstream relation classification. Moreover, both $\mathcal{L}_{\text{mask}}$ and $\mathcal{L}_{\text{div}}$ contribute effectively to enhancing model performance. The former provides fine-grained structural guidance over the temporal intervals, while the latter encourages the predicted interval masks to capture diverse temporal patterns. Combining these three losses can yield better performance.

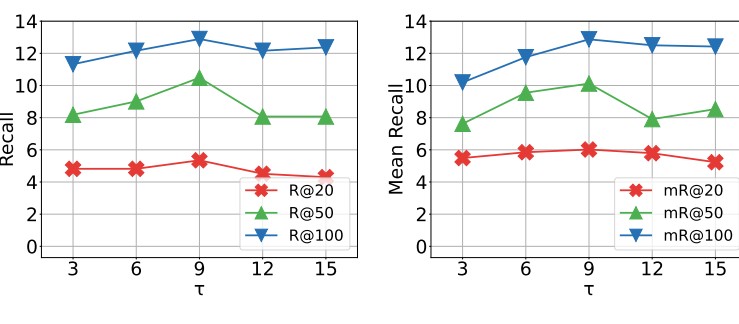

Figure 6: Ablation of the Gaussian mask variance parameter $\tau$ for TFNet.

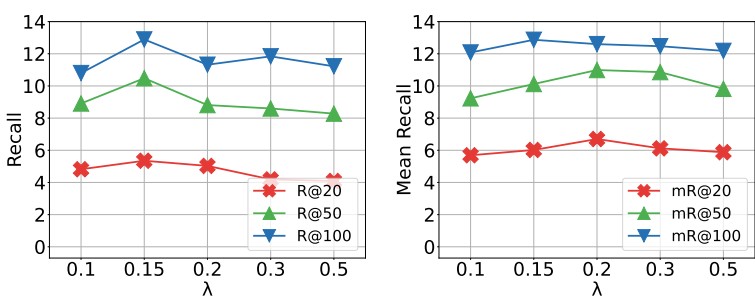

Figure 7: Ablation of the diversity weight $\lambda$ for TFNet.

### A.1.3 EFFECT OF THE GAUSSIAN MASK SHARPNESS.

We investigate how the shape of the Gaussian mask affects relation modeling. The shape is controlled by the variance parameter $\tau$ in Equation (7), where a larger $\tau$ results in a sharper shape. We vary $\tau$ within the set $\{3, 6, 9, 12, 15\}$ and report the corresponding results in Figure 6. As shown, the model achieves the best performance when $\tau = 9$, suggesting a balance between precise temporal localization and tolerance to slight misalignment. Both overly sharp and overly flat masks degrade performance, highlighting the importance of appropriately modeling the temporal structure of relation intervals.

### A.1.4 COMPARISON OF SCORE AGGREGATION STRATEGIES.

We conduct an ablation study to compare two score aggregation strategies used to compute the final relation score for retrieval in our Score-based Merging Strategy. Specifically, we evaluate a variant of our method where the average score in Equation (14) is replaced by the maximum score. As shown in Table 7, the average score achieves better performance on most metrics. These re-

Table 7: Ablation study on score aggregation strategies used in the Score-based Merging Strategy.

| Score Strategy | R/mR@20 | R/mR@50 | R/mR@100 |
|---|---|---|---|
| Max Score | 4.82 / 5.16 | 8.81 / **10.22** | 11.32 / 12.34 |
| Ave Score | **5.35 / 6.02** | **10.48** / 10.12 | **12.89 / 12.87** |

sults demonstrate that averaging provides a more robust and balanced estimation of relation confidence, as it effectively integrates evidence from multiple intervals. In contrast, the maximum score tends to overemphasize single predictions, making the final relation score more sensitive to noise.

### A.1.5 EFFECT OF DIVERSITY DEGREE FOR GAUSSIAN MASKS.

We conduct an ablation study to investigate the effect of the diversity regularization weight $\lambda$ in Equation (9), which controls the strength of the diversity regularization applied to the predicted Gaussian masks. We vary $\lambda$ in the range $\{0.1, 0.15, 0.2, 0.3, 0.5\}$ and report the results in Figure 7. As shown, performance improves as $\lambda$ increases from 0.1 to 0.15 or 0.2, then declines when $\lambda$ becomes larger. These results suggest that an appropriate $\lambda$ promotes diversity among temporal queries, thereby enhancing the ability of the model to cover varied relation patterns.

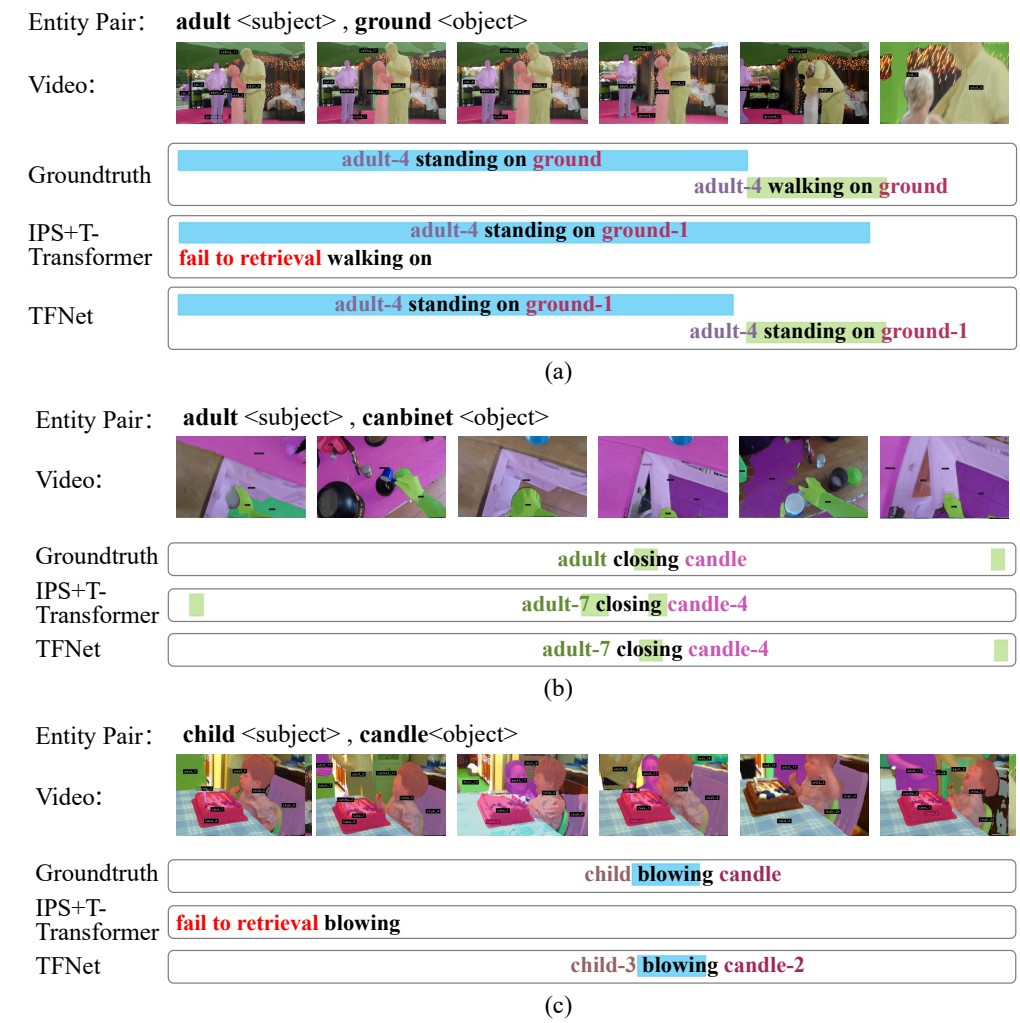

Figure 8: Additional comparisons between our TFNet with IPS+T-Transformer (Yang et al. 2023)

## A.2 MORE VISUALIZATION RESULTS

Figure 8 shows some qualitative examples. As shown in Figure 8(a), the relation between *adult_4* and *ground_1* changes over time from "standing on" to "walking on". While the prior method only detects the static "standing on" relation throughout the sequence, our TFNet accurately captures the transition and localizes both relations with precise temporal boundaries, demonstrating its capability in handling dynamic relations in PVSG. In Figure 8(b), our approach successfully detects all interaction intervals with accurate time, highlighting that TFNet can better handle complex temporal dynamics and recurrent relations. In Figure 8(c), TFNet again shows robustness by successfully retrieving the fine-grained relation *blowing*. This example demonstrates that our method effectively captures subtle and transient interactions. Notably, our approach performs well in both third-person and egocentric views, reflecting its robustness across diverse visual perspectives.

## A.3 LLM USAGE

Large Language Models (LLMs) were used to aid in the writing and polishing of the manuscript. Specifically, we used an LLM to assist in refining the language, improving readability, and ensuring clarity in various sections of the paper. The model helped with tasks such as sentence rephrasing, grammar checking, and enhancing the overall flow of the text.

It is important to note that the LLM was not involved in the ideation, research methodology, or experimental design. All research concepts, ideas, and analyses were developed and conducted by the authors. The contributions of the LLM were solely focused on improving the linguistic quality of the paper, with no involvement in the scientific content or data analysis.

The authors take full responsibility for the content of the manuscript, including any text generated or polished by the LLM. We have ensured that the LLM-generated text adheres to ethical guidelines and does not contribute to plagiarism or scientific misconduct.

