# OpenReview forum: "Temporally Aligned Relation Modeling for Panoptic Video Scene Graph Generation"
_ICLR.cc/2026/Conference — Submitted to ICLR 2026_

### Official Review · Reviewer_Ls1m · 2025-10-19

**Soundness:** 3
**Presentation:** 3
**Contribution:** 2
**Rating:** 6
**Confidence:** 4

**Summary:**

This paper propose TFNet，a new framework for video scene graph generation, particularly for accurately localize and classify temporal relations in videos. The core innovation is to introduce the temporal query mechanism to capture the temporal relations in videos.

**Strengths:**

The writing is clear and the method is reasonable. The results are good.

**Weaknesses:**

The localization-then-classification framework has been previously explored, and video object localization, classification, and feature construction are based on existing methods. The primary contribution of this paper is the introduction of the temporal query mechanism to achieve temporal relation localization followed by classification. Overall, I am not sure whether this innovation is sufficient.

Regarding why such a straightforward method performs so well, the ablation study lacks a more detailed analysis to specify which components contribute to the performance improvement.

**Questions:**

1. Could the physical significance of the temporary query be further explained?
2. In Table 4, TFNet shows substantial performance improvement compared to other methods. Could a more detailed ablation study be conducted to specify how this performance gain is accumulated?
3. In line 206, how is the hyperparameter T set in the experiments? What is the sampling frequency for video frames, and how does it affect the final performance?
4. How efficient is this method in practical inference? It is suggested to provide some video demos for a more intuitive demonstration of the effects in the furture.
5. Have the authors explored how current multimodal large models perform on video SGG? What are the differences in performance and inference efficiency?
6. It is recommended to include the following references:
- Wang, G., Li, Z., Chen, Q. and Liu, Y., 2024. Oed: Towards one-stage end-to-end dynamic scene graph generation. In Proceedings of the IEEE/CVF conference on computer vision and pattern recognition (pp. 27938-27947).
- Zhang, Y., Pan, Y., Yao, T., Huang, R., Mei, T. and Chen, C.W., 2023. End-to-end video scene graph generation with temporal propagation transformer. IEEE Transactions on Multimedia, 26, pp.1613-1625.
- Wu, S., Fei, H. and Chua, T.S., 2025. Universal scene graph generation. In Proceedings of the Computer Vision and Pattern Recognition Conference (pp. 14158-14168).

---

### Official Review · Reviewer_cVA1 · 2025-10-31

**Soundness:** 3
**Presentation:** 3
**Contribution:** 2
**Rating:** 4
**Confidence:** 4

**Summary:**

The paper tackles Scene Graph Generation by predicting interaction intervals as 2D Gaussian mask, differentiating itself from perframe prediction of previous works.

**Strengths:**

The development of new Gaussian mask is clear and well-motivated.

**Weaknesses:**

- The paper should more clearly differentiate and highlight the proposed interval-based approach from the per-frame prediction methods in prior work. A substantial portion of the methodology section reiterates well-established practices in identification and localization; while this may benefit readers new to the field, it appears somewhat redundant relative to the paper’s novel contributions.

- While the method is empirically driven, there is little theoretical insight into why the Gaussian formulation should better capture temporal intervals or uncertainty. This could be better illustrated through equations and a discussion of the mathematical properties that capture temporal and context-aware characteristics.

- The main contribution appears to be the introduction of a Gaussian mask for interval prediction, which seems incremental, as several prior works have also explored holistic approaches to the task [1].

[1] Nguyen, et al. Hig: Hierarchical interlacement graph approach to scene graph generation in video understanding. CVPR 2024.

**Questions:**

See weaknesses.

---

### Official Review · Reviewer_G8nw · 2025-11-02

**Soundness:** 3
**Presentation:** 3
**Contribution:** 3
**Rating:** 4
**Confidence:** 5

**Summary:**

This paper proposes TempFocusNet (TFNet), a framework for Panoptic Video Scene Graph Generation (PVSG). The core contribution is a "localize-then-recognize" paradigm: it first generates candidate relation intervals, represented as Gaussian masks, and then uses these masks to guide a temporal attention mechanism for focused relation classification within the identified intervals. The authors argue that this approach avoids the noise introduced by modeling relations over the entire video. The method is reported to achieve state-of-the-art performance on the OpenPVSG and ImageNet-VidVRD benchmarks.

**Strengths:**

1. Clear and Well-Structured Methodology:The paper presents a clear pipeline. The description of constructing entity tubes, using temporal queries to generate Gaussian masks, and applying temporal focus attention is logically sound and easy to follow.

2. Strong Empirical Results:The reported state-of-the-art performance on two standard benchmarks (OpenPVSG and ImageNet-VidVRD) is a strong point and demonstrates the practical effectiveness of the proposed approach.

**Weaknesses:**

1. Limited Conceptual Novelty:The central idea of "first localize the interval, then recognize the relation" is not a new paradigm in Video Scene Graph Generation (VidSGG) or related video understanding tasks.Similar "detect-then-classify" or proposal-based strategies have been explored in prior work for temporal action localization, video relation detection, and indeed, in some existing VidSGG methods. The paper's introduction does not adequately acknowledge this lineage or precisely define how TFNet's instantiation of this idea is fundamentally different or novel compared to these existing approaches.

2. Weak Motivational Presentation:The motivation, while valid, is presented as a binary choice: existing methods use the "entire video sequence" while TFNet uses "focused intervals." This is an oversimplification. Many modern video methods already employ attention mechanisms or temporal convolutions that can adaptively focus on relevant parts of a sequence. A more compelling motivation would involve a critical analysis of whycurrent adaptive methods still fail and how the explicit, hard-localization step of TFNet provides a unique solution.

**Questions:**

Qualitative Analysis:The paper would be significantly strengthened by strong qualitative results. Please include visualizations that show:
Examples where your method correctly localizes a relation interval that a strong baseline model misses or misclassifies.

---

### Meta-Review · Area_Chair_27Vg · 2026-01-08

**Summary:**

The reviewers agree that the paper presents a clear and well-structured framework for panoptic video scene graph generation, with strong empirical results on established benchmarks. The use of temporally localized relation intervals—implemented via Gaussian masks and temporal queries—to focus relation classification is considered reasonable, and the overall presentation is clear and accessible. Multiple reviewers acknowledge that the method is practically effective and that the reported results are competitive.

However, reviewers consistently raise concerns about limited novelty, noting that the localization-then-classification paradigm and many architectural components are well explored in prior VidSGG and video understanding work. The main contribution—Gaussian interval masks or temporal queries—is viewed as incremental, and the paper does not sufficiently distinguish itself from existing holistic or adaptive temporal modeling approaches. Across reviews, there is a shared request for stronger motivation, deeper ablation and theoretical analysis to explain the source of performance gains, and more qualitative and efficiency-focused evaluation to better justify the method’s impact.

There are no rebuttals. Thus, the AC decides to reject this work.

**Reviewer Scores:**

no

---

### Decision · Program_Chairs · 2026-01-26

Reject